# Medial Patellofemoral Ligament Repair with Suture Tape Augmentation Can Yield Good Midterm Clinical Outcomes Regardless of Skeletal Maturity and Joint Laxity

**DOI:** 10.3390/biomimetics10010065

**Published:** 2025-01-18

**Authors:** Shinichiro Takada, Hirotaka Nakashima, Keisuke Nakayama, Soshi Uchida

**Affiliations:** Department of Orthopedic Surgery, Wakamatsu Hospital of University of Occupational and Environmental and Health, 1-17-1, Hamamachi, Wakamatsu, Kitakyushu-city 808-1264, Fukuoka, Japan; shin-takada@med.uoeh-u.ac.jp (S.T.); hirotakanakashima58@gmail.com (H.N.); comeonkeisuke@gmail.com (K.N.)

**Keywords:** MPFL repair with suture tape augmentation, polyethylene, patella dislocation, patella instability

## Abstract

While several studies have reported short-term clinical outcomes after medial patellofemoral ligament (MPFL) repair with suture tape augmentation, there is still a dearth of knowledge regarding midterm clinical outcomes. This study aimed to evaluate the midterm clinical outcomes of MPFL repair with suture tape augmentation in patients with patellar dislocation. We retrospectively reviewed the clinical records of patients who underwent MPFL repair with suture tape augmentation for at least one episode of patellar dislocation between 2015 and 2020. Patient-reported clinical outcomes (PROs) were evaluated via the International Knee Documentation Committee (IKDC) score and the knee injury osteoarthritis outcome score (KOOS). In total, 17 knees (4 males and 13 females) who underwent MPFL repair with suture tape augmentation with a mean follow-up of 54.6 ± 19.5 months were included in this study. PROs significantly improved from preoperatively to the final follow-up (IKDC score: 50.7 ± 26.6 vs. 88.8 ± 13.0, *p* < 0.001; KOOS: 68.8 ± 23.3 vs. 91.2 ± 8.4, *p* = 0.011) without reducing the patient’s activity level at the final follow-up (UCLA AS score: 7.9 ± 2.4 at preinjury vs. 7.9 ± 2.2 at the final follow-up, *p* = 0.655). Subgroup analysis revealed good postoperative outcomes, regardless of the patient’s skeletal maturity or the presence or absence of generalized laxity. In conclusion, MPFL repair with suture tape augmentation is a safe and effective treatment for midterm follow-up.

## 1. Introduction

Medial patellofemoral ligament (MPFL) reconstruction is widely considered the standard procedure for treating recurrent patellar instability because of its high success rate and low risk of recurrent instability [1]. A systematic review revealed that the pooled risk of recurrent instability after MPFL reconstruction was 1.2% [2]. Kruckeberg et al. demonstrated that, compared with MPFL repair, MPFL reconstruction yields lower redislocation rates and better functional outcomes [3]. However, it is associated with a significant complication rate. Studies report overall complication rates ranging from 0% to 32.3% [4]. A systematic review reported a 26.1% complication rate in 629 knees, including donor site morbidity and patellar fractures, which are associated with graft harvesting [5]. In particular, patellar fractures account for approximately 8% of complications in MPFL reconstruction and are associated primarily with specific surgical techniques and tunnel diameters [4]. Recently, several studies have demonstrated that MPFL surgery using suture tape, an ultrahigh-strength artificial ligament composed of long chains of ultrahigh-molecular-weight polyethylene augmentation, may be a valuable alternative for restoring patellar stability, avoiding complications associated with autograft harvest and bone tunnels [6,7]. The design of polyethylene-based suture tape is inspired by the hierarchical structure and biomechanical properties of natural ligaments and tendons, which exhibit exceptional tensile strength, flexibility, and durability due to their collagen fiber alignment and composite structure [8]. Polyethylene’s high molecular weight and crystallinity mimic the load-bearing capacity of collagen fibrils, while its smooth surface reduces friction, reflecting the low-friction gliding properties of synovial tissues. These bioinspired characteristics enable the suture tape to adapt to dynamic stresses, enhance biocompatibility, and provide durability and flexibility, making it ideal for ligament repair applications such as MPFL reconstruction [8,9]. The tape is supposed to act as a secondary stabilizer and protect the native ligament during its biological healing. A biomechanical study revealed that repair with suture tape augmentation provides similar joint kinematics and contact pressures to those of MPFL reconstruction [10,11].

Suture tape has proven not only to be safe and effective but also to produce comparable patient-reported outcomes to those of autologous tendon reconstruction [12]. Several studies have described favorable clinical outcomes after MPFL surgery via suture tape augmentation [13,14,15], but there is still a dearth of knowledge regarding midterm clinical outcomes. This study aimed to evaluate the midterm clinical outcomes of MPFL repair with suture tape augmentation in patients with patellar dislocation.

## 2. Materials and Methods

### 2.1. Subjects

This research was approved by the IRB of the authors’ affiliated institutions (approval number: ∗UOEHCRB20-171), and all study subjects provided informed consent.

We retrospectively reviewed the clinical records of patients who underwent MPFL repair with suture tape augmentation between 2015 and 2020. The indications for the current procedure were patients who had at least one episode of patellar dislocation without trochlear dysplasia (sulcus angle > 150 degrees on MRI) or malalignment (tibial tubercle-trochlear groove (TT-TG) distance > 20 mm) because of the need for other procedures. The clinical records included patient demographics, radiographic variables, and clinical outcomes. 

The clinical records included patient demographics, radiographic variables, and clinical outcomes. Patients with short follow-up periods (within 2 years) were excluded.

### 2.2. Surgical Technique

The surgical technique used for MPFL repair with suture tape augmentation is shown in Figure 1. The patient was placed in the supine position under general anesthesia, and standard diagnostic arthroscopy was performed to evaluate cartilage damage of the patella and trochlea. First, a 3–5 cm arcuate incision was made on the medial edge of the patella. The superficial attachment of the vastus medialis obliquely was incised, and the MPFL-capsule complex was detached from the patella. If there was a bony avulsion fragment, it was removed via an electric scalpel. Second, a 3 cm longitudinal skin incision was made just above the medial femoral condyle. The saphenous nerve should be isolated if it is detected. The femoral-side footprint of the MPFL was confirmed under fluoroscopy, and a 2.4 mm K-wire was inserted into Schöttle’s point [16,17] and overdrilled for SwiveLock (Arthrex, Naples, FL, USA) fixation (Figure 1A). A double-folded polyester high-strength suture tape (FiberTape; Arthrex) was passed through the eyelet of a knotless anchor (SwiveLock; Arthrex) and fixed to the femoral footprint. Suture tape was removed through the MPFL-capsule complex substance to avoid infrapatellar injury of saphenous N. via Arthropierce (Figure 1B). Third, two K-wires were inserted at the 1:00 and 3:00 positions, taking care not to interfere with each other and aiming to cover the medial patellofemoral complex (MPFC), including the MPFL and medial quadriceps tendon femoral ligament (MQTFL) [18] (Figure 1C). After isometrisity was confirmed, the suture tapes were fixed to the patella via SwiveLock (Arthrex, Naples, FL, USA) anchors at 60 degrees of knee flexion while keeping the patella at the center of the trochlear groove. Finally, the MPFL-capsule complex was repaired via the ends of the suture tapes and the SwiveLock (Arthrex, Naples, FL, USA) sutures via the lasso loop technique (Figure 1D).

### 2.3. Postoperative Rehabilitation

Gentle passive range-of-motion exercise was initiated immediately after surgery. There was no restriction of ROM after surgery. Progressive partial weight-bearing with crutches was immediately allowed. After 4 weeks, full weight-bearing exercise was permitted as tolerated. Open kinetic chain exercise was started at 3 weeks and closed kinetic chain exercise was started at 10 weeks after surgery. Deep squatting and jogging were allowed at 3 months after surgery, and a return to sports was allowed after 6 months.

### 2.4. Clinical Outcome Evaluation

Clinical outcomes were evaluated via patient-reported clinical outcomes (PROs), the UCLA activity scale, and complications. PROs were evaluated via the International Knee Documentation Committee (IKDC) score and the Knee Injury Osteoarthritis Outcome (KOOS) score preoperatively, 6 months postoperatively, and at the final follow-up. The patient acceptable symptomatic state (PASS) of the IKDC score and KOOS were assessed via a previous study. [19] Thus, the PASS score of the IKDC was defined as 65.5. The PASS scores of the KOOS for Symptom, Pain, ADL, Sport, and QOL were 80.4, 84.7, 99.3, 57.5, and 53.1, respectively. The UCLA activity scale was evaluated preinjury, preoperatively, and at the final follow-up. Complications, including infection and redislocation, were also evaluated.

Subgroup analysis was also performed regarding general joint laxity and skeletal maturity.

### 2.5. Imaging Evaluation

The Insall–Salvati ratio was evaluated via the lateral view. The sulcus angle and congruence angle were evaluated via the skyline (axial) view [20]. Skeletal immaturity was defined by radiographic epiphysis via the anterior–posterior view and lateral view. The TT–TG distance and bony fragment were evaluated via computed tomography.

### 2.6. Statistical Analysis

The Wilcoxon rank test was used to compare the PRO and UCLA activity scales after surgery. The Mann–Whitney U test was used to compare the skeletal immature and mature groups, except for qualitative variables, which were compared via the chi-square test or Fisher’s exact test. All the statistical analyses were performed via SPSS software (version 24; SPSS Inc., Chicago, IL, USA).

## 3. Results

### 3.1. Patient Recruitment

During this period, 25 knees underwent MPFL repair with suture tape augmentation, and 8 knees were excluded from this study because of the short follow-up period (follow-up rate of 68.0%). Finally, 17 knees (4 males and 13 females) were assessed. The mean age at surgery was 18.0 ± 5.8 years, and the mean BMI was 23.1 ± 4.0 kg/m^2^. The mean follow-up was 54.6 ± 19.5 months. Overall, 6 patients (3 males and 3 females) were skeletally immature, and 11 patients (1 male and 11 females) were skeletally mature. Demographics and preoperative radiographic variables are summarized in Table 1.

### 3.2. Clinical Outcomes

The mean IKDC score significantly improved from 50.7 ± 26.6 preoperatively to 79.1 ± 18.1 (*p* = 0.025) at 6 months postoperatively and to 88.8 ± 13.0 (*p* < 0.001) at the final follow-up (Figure 2). At the final follow-up, 94.1% (16/17 patients) achieved PASS at the IKDC.

Similarly, the mean KOOS Symptom score significantly improved postoperatively. The mean KOOS Symptom score significantly improved from 68.8 ± 23.3 preoperatively to 88.5 ± 10.0 (*p* = 0.018) at 6 months postoperatively and 91.2 ± 8.4 (*p* = 0.011) at the final follow-up (Figure 3A). At the final follow-up, 88.2% (15/17 patients) achieved PASS in the KOOS Symptom score. The mean KOOS Pain score significantly improved from 68.5 ± 25.7 preoperatively to 88.2 ± 13.9 (*p* = 0.031) at 6 months postoperatively and 91.4 ± 8.8 (*p* = 0.002) at the final follow-up (Figure 3B). At the final follow-up, 76.5% (13/17 patients) achieved PASS in the KOOS Pain. The mean KOOS ADL score significantly improved from 77.9 ± 21.6 preoperatively to 96.6 ± 4.5 (*p* = 0.047) at 6 months postoperatively and 98.9 ± 1.8 (*p* = 0.003) at the final follow-up (Figure 3C). At the final follow-up, 100.0% (17/17 patients) achieved PASS in the KOOS ADL. The mean KOOS Sport score significantly improved from 43.0 ± 36.7 preoperatively to 77.3 ± 21.2 (*p* = 0.002) at 6 months postoperatively and 91.5 ± 8.6 (*p* < 0.001) at the final follow-up (Figure 3D). At the final follow-up, 100.0% (17/17 patients) achieved PASS in the KOOS Sport score. The mean KOOS QOL significantly improved from 46.7 ± 28.3 preoperatively to 73.6 ± 19.6 (*p* = 0.004) at 6 months postoperatively and 82.0 ± 14.8 (*p* = 0.002) at the final follow-up (Figure 3E). At the final follow-up, 100.0% (17/17 patients) achieved PASS in the KOOS QOL.

The mean UCLA activity scales were 7.9 ± 2.4 at preinjury, 4.8 ± 3.7 at preinjury, and 7.9 ± 2.2 at the final follow-up (Figure 4). Compared with that before injury, the UCLA activity scale at the final follow-up significantly improved (*p* = 0.007), and the UCLA activity scale at the final follow-up was comparable to that at preinjury (*p* = 0.655).

The results of the subgroup analysis are presented in Table 2 and Table 3. There were no significant differences in PROs between the two groups in terms of general joint laxity or skeletal maturity.

Redislocation was observed in 1 patient at 14 months after initial surgery because of a new trauma episode. There were no cases of infection.

## 4. Discussion

The most important finding of this study was that MPFL repair with suture tape augmentation could achieve favorable clinical outcomes even in the midterm. Significant improvement was confirmed in terms of PROs at 6 months after surgery, and this improvement was maintained for more than 2 years. The patient’s activity level also returned to preoperative levels. Furthermore, there was significant improvement even in skeletally immature patients, which was equivalent to that observed in skeletally mature patients. Moreover, the results of the present study demonstrated that patients who experienced generalized joint laxity could also be successfully treated via MPFL repair with suture tape augmentation.

Previous studies reported several technical notes of MPFL surgery with suture tape augmentation [6,7,13,21]. Some studies were entitled ‘MPFL reconstruction via suture tape augmentation’, and others were entitled ‘MPFL repair via suture tape augmentation’. There is no clear standard for distinguishing between ‘repair’ and ‘reconstruction’ with suture tape augmentation. Among these studies, the difference in the surgical technique used was the direction of suture tape fixation. Some reports have described a method of inserting a suture tape on the patella side, pulling it to the femur side, and fixing it to the femur side as MPFL reconstruction via suture tape [7,13]. Hopper et al. reported an MPFL repair with the suture tape augmentation method in which a suture tape is inserted on the femoral side and then fixed to the patella side after primary repair on the patella side [6]. Our surgical method was similar to Hopper’s method; we placed the suture tape on the femoral side, pulled it out to the patellar side, and fixed it to the patella. After that, we performed primary MPFL repair via the remaining sutures of the patellar side anchor. While MPFL injury patterns vary, studies consistently show a higher prevalence of patellar-side MPFL injuries. Askenberger et al. reported that 66% of injuries involved patellar attachment in skeletally immature patients [22]. Kluczynski et al. conducted a systematic review including 35 articles (2558 patients) and concluded that MPFL injuries at the patella were most prevalent overall and in children and adolescents [23]. We believe that our current technique is a reasonable procedure for MPFL repair with suture tape augmentation.

Several studies have reported the clinical outcomes after MPFL repair or reconstruction with suture tape augmentation. Xu et al. reported good functional outcomes via the Lysholm score, the SF-12 score, and the Tegner score in a 12-month follow-up study [13]. They also reported a significant improvement in radiological findings and concluded that MPFL reconstruction via suture tape has good clinical results and can improve the stability of the knee. Sasaki et al. reported good clinical outcomes in terms of the KOOS in a 2-year follow-up study (minimum 1 year). They concluded that all the KOOS subscale scores improved regardless of the preoperative patellar height or TT–TG distance. In the present study, we revealed favorable patient-reported clinical outcome scores after MPFL repair with suture tape augmentation at a mean follow-up of 54 months (minimum 2 years). In this study, PROs were significantly improved at 6 months after surgery, and this improvement was maintained at the final follow-up regardless of general joint laxity, skeletal maturity, or bony fragmentation. Notably, the IKDC score and KOOS subscale scores were high for the PASS achievement rate, especially for the ADL, Sports, and QOL subscales. This means that almost all patients were satisfied with their postoperative outcomes.

In skeletally immature patients, there is concern about disruption of the physis resulting in growth disturbance or arrest after MPFL reconstruction. Seitlinger et al. reported a case of partial posterior physeal growth arrest and subsequent flexion deformity of the distal femur three years after MPFL reconstruction in a skeletally immature patient [24]. Studies have shown comparable functional outcomes and redislocation rates between skeletally immature and mature groups [25]. However, skeletally immature patients may have a greater risk of subsequent ipsilateral injury [25,26] and lower satisfaction rates [26] than skeletally mature patients do. Hobson et al. reported that MPFL reconstruction with tape augmentation significantly reduced subsequent ipsilateral knee injuries but that there was no significant difference in the incidence of PROs or recurrent dislocations compared with nonaugmented MPFL reconstruction in skeletally mature adolescents [26]; no comparative studies regarding the effects of skeletal maturity on the clinical outcomes after MPFL repair with suture tape augmentation surgery exist. In the present study, we found significant improvement in PROs in both the skeletally immature and mature groups, and there was no significant difference after MPFL repair with tape augmentation. Further clinical comparative studies may be needed to elucidate the effectiveness of this procedure for pediatric patients.

Generalized joint laxity (GJL) has been identified as a risk factor for poor outcomes following ligament reconstruction. Studies have shown that GJL is associated with inferior results following anterior cruciate ligament (ACL) reconstruction [27]. Long-term follow-up studies revealed that patients with GJL had higher rates of graft rupture and contralateral ACL rupture than those without GJL [28]. These findings suggest that GJL should be considered a risk factor for poor outcomes and that careful planning is necessary when performing ligament reconstructions in this patient population. Regarding the relationship between patellar instability and GJL, Heighes et al. compared 104 patients with patellar dislocation to 110 patients without dislocation [29]. They reported that the prevalence of GJL was six times greater in the patella dislocation group than in the other groups in a systematic review [29]. Parikh et al. reported a high revision rate (38.3%) in patients with Ehlers–Danlos syndrome who underwent MPFL reconstruction for patellar instability [30]. Galán-Olleros et al. reported 29 knee case series including GJL patients who underwent MPFL reconstruction with suture tape augmentation [31]. Their study demonstrated good functional outcomes and a high satisfaction rate (79.3%) overall, but there is still a dearth of knowledge regarding the effectiveness of MPFL surgery with suture tape augmentation for GJL patients. Our current study is the first to perform a comparative analysis between patients with GJL and those without GJL. PROs significantly improved regardless of GJL after MPFL repair with tape augmentation.

This study has several limitations. First, there was no control group, and the sample size was relatively small. A larger number of cases would be needed to make current subgroup analysis more meaningful information. To evaluate the effectiveness of the current procedure, it would be better to compare it with isolated MPFL repair or MPFL reconstruction. Furthermore, while we utilized the KOOS to evaluate clinical outcomes, it is possible that the KOOS may not be ideal as the sole PROM for assessing patellar instability. The knee injury and osteoarthritis outcome score for the patellofemoral pain and osteoarthritis (KOOS-PF) subscale was specifically developed for patellofemoral pain and osteoarthritis and has shown good reliability, validity, and responsiveness [32]. A systematic review highlighted the KOOS-PF as the only PROM with sufficient content validity for patellofemoral pain, despite low-quality evidence supporting its use [33]. While the KOOS and its subscales remain valuable tools, condition-specific PROMs like the KOOS-PF may offer a more targeted assessment for patellofemoral issues and patellar instability.

Second, we did not evaluate trochlear dysplastic patients or abnormal alignment patients. These findings suggest that the findings of this study may not fully represent broader populations or varying clinical scenarios of patella instability patients. Third, although this was a mid-term outcome study over more than 4 years, further long-term outcomes are needed. Although a biomechanical study revealed that there was no difference between MPFL repair with suture tape augmentation and reconstruction with a hamstring graft in terms of patellofemoral (PF) contact pressure and joint kinematics at time zero [11], researchers have not determined whether tape augmentation is related to the future onset of cartilage damage to the PF joint or subsequent knee damage in the long term in clinical practice. Addressing these limitations through larger-scale multicenter studies with extended follow-up will be essential to solidify the evidence base for MPFL repair with suture tape augmentation and optimize its clinical application.

## 5. Conclusions

MPFL repair with suture tape augmentation is a safe and effective treatment regardless of the patient’s skeletal maturity or the presence or absence of generalized laxity in midterm follow-up.

## Figures and Tables

**Figure 1 biomimetics-10-00065-f001:**
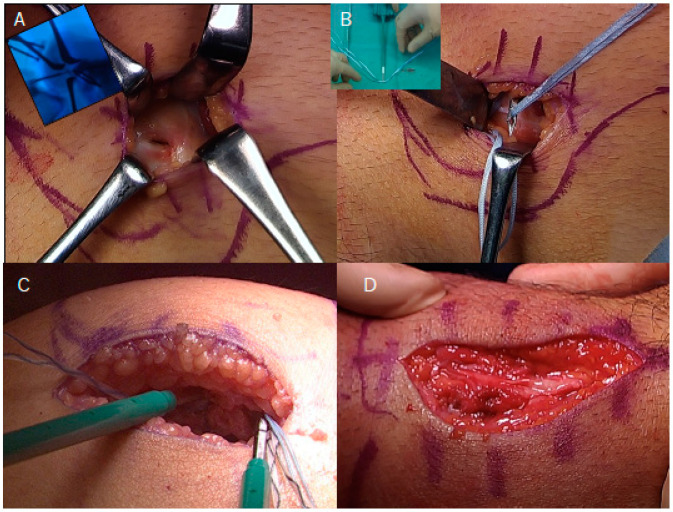
Intraoperative findings of the right knee viewing from the medial region. The left side of the photograph is distal, and the right side is proximal. (**A**) Schöttle’s point was identified under fluoroscopic guidance, and an anchor was inserted precisely at the femoral insertion point. (**B**) Careful dissection through the minimal incision allowed for the MPFL-capsule complex to be visualized, and suture tape was passed through the MPFL substance using an Arthropierce device. (**C**) Two SwiveLock (Arthrex, Naples, FL, USA) anchors were placed at the 1:00 and 3:00 positions on the patella to cover the medial patellofemoral complex (MPFC) while maintaining isometry. (**D**) The MPFL-capsule complex was repaired using the lasso loop technique with suture tape and SwiveLock (Arthrex) anchors, ensuring proper tension and alignment within the trochlear groove.

**Figure 2 biomimetics-10-00065-f002:**
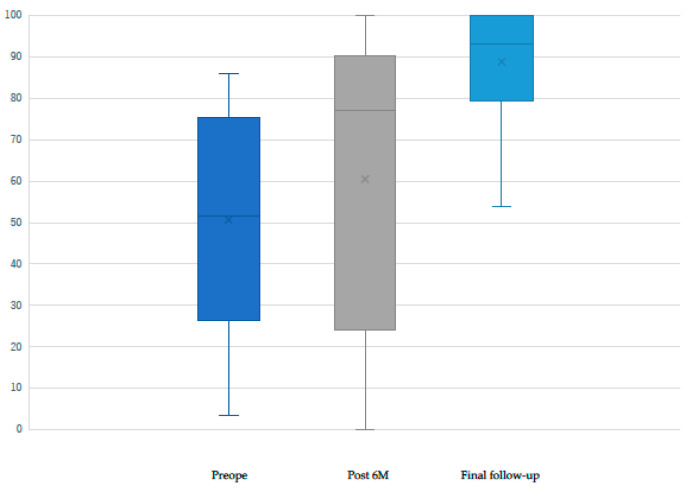
Preoperative and postoperative IKDC scores. The IKDC score significantly improved after surgery.

**Figure 3 biomimetics-10-00065-f003:**
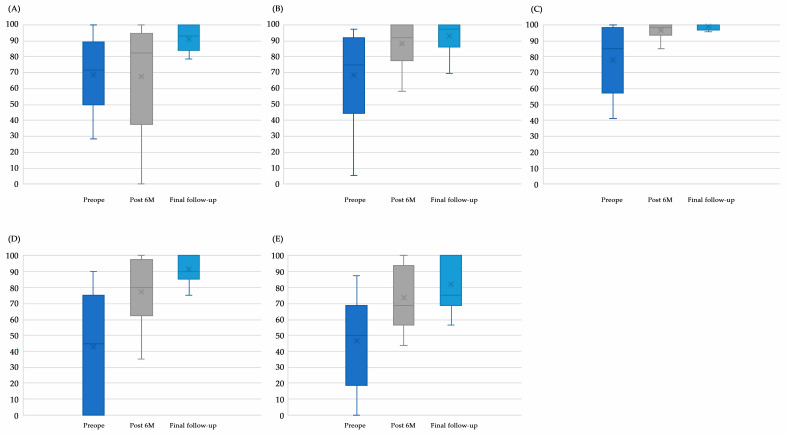
Preoperative and postoperative KOOS. The scores of the KOOS subscales significantly improved after surgery. (**A**): Symptom, (**B**): Pain, (**C**): ADL, (**D**): Sport, (**E**): QOL.

**Figure 4 biomimetics-10-00065-f004:**
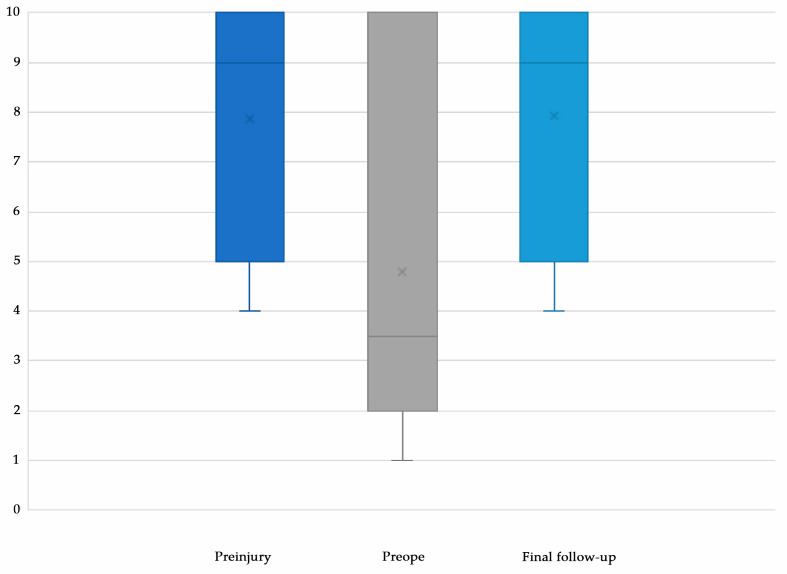
UCLA activity scale. The UCLA activity scale significantly improved after surgery, and the UCLA activity scale at the final follow-up was comparable to that before surgery.

**Table 1 biomimetics-10-00065-t001:** Patient demographics and preoperative radiographic variables. The data shown are the means ± SDs and standard deviations.

	Overall (N = 17)
Age (years)	18.0 ± 5.8
Sex (Male: Female)	4: 13
BMI (kg/m^2^)	23.1 ± 4.0
Follow-up (months)	54.6 ± 19.5
General Joint laxity (+ : −)	7: 10
Skeletal immature: mature	6: 11
Sulcus angle (degree)	144.5 ± 8.2
Congruence angle (degree)	16.5 ± 16.8
Insall-Salvati	1.27 ± 0.2
TT-TG distance (mm)	17.2 ± 2.9
Bony fragment(+ : −)	10: 7

**Table 2 biomimetics-10-00065-t002:** Comparison of PROs between the general joint laxity group and the no general joint laxity group.

	General Joint Laxity Group (n = 7)	No General Joint Laxity Group (n = 10)	*p*-Value
IKDC preope	59.9 ± 26.0	44.3 ± 26.4	0.364
IKDC post 6 months	81.1 ± 22.0	76.8 ± 13.9	0.445
IKDC final follow-up	90.6 ± 10.5	86.2 ± 16.4	0.536
KOOS Symptom preope	75.7 ± 24.8	65.4 ± 23.0	0.440
KOOS Symptom post 6 months	86.9 ± 11.4	89.8 ± 9.3	0.731
KOOS Symptom final follow-up	91.8 ± 7.4	90.7 ± 9.4	0.887
KOOS Pain preope	79.4 ± 9.8	63.1 ± 29.7	0.440
KOOS Pain post 6 months	84.3 ± 17.4	91.7 ± 10.4	0.445
KOOS Pain final follow-up	94.0 ± 5.9	92.2 ± 10.6	0.887
KOOS ADL preope	87.9 ± 17.7	72.9 ± 22.4	0.310
KOOS ADL post 6 months	97.8 ± 3.0	95.6 ± 5.4	0.628
KOOS ADL final follow-up	98.5 ± 1.9	99.1 ± 1.9	0.601
KOOS Sport preope	60.0 ± 34.1	34.5 ± 36.5	0.440
KOOS Sport post 6 months	78.3 ± 23.2	76.4 ± 21.2	0.945
KOOS Sport final follow-up	90.7 ± 7.3	92.0 ± 9.8	0.740
KOOS QOL preope	66.3 ± 13.0	36.9 ± 29.2	0.075
KOOS QOL post 6 months	75.0 ± 15.8	72.3 ± 23.6	0.731
KOOS QOL final follow-up	78.6 ± 12.9	84.4 ± 16.2	0.364

**Table 3 biomimetics-10-00065-t003:** Comparison of PROs between the immature skeletal group and the mature skeletal group.

	Skeletal Immature Group (n = 6)	Skeletal Mature Group (n = 11)	*p*-Value
IKDC preope	49.6 ± 35.5	51.3 ± 22.3	0.884
IKDC post 6 months	81.8 ± 18.2	77.5 ± 19.1	0.724
IKDC final follow-up	87.2 ± 9.0	89.7 ± 15.0	0.256
KOOS Symptom preope	73.8 ± 18.4	65.5 ± 26.5	0.607
KOOS Symptom post 6 months	90.0 ± 9.2	87.5 ± 11.0	0.943
KOOS Symptom final follow-up	87.5 ± 7.4	93.2 ± 8.5	0.149
KOOS Pain preope	62.0 ± 35.0	72.8 ± 18.4	0.776
KOOS Pain post 6 months	92.2 ± 10.1	85.8 ± 16.0	0.724
KOOS Pain final follow-up	89.8 ± 11.6	94.7 ± 6.9	0.301
KOOS ADL preope	73.8 ± 25.9	80.7 ± 19.3	0.955
KOOS ADL post 6 months	97.1 ± 3.1	96.3 ± 5.3	0.943
KOOS ADL final follow-up	98.8 ± 2.0	98.9 ± 1.9	0.884
KOOS Sport preope	39.2 ± 39.8	45.6 ± 36.7	0.529
KOOS Sport post 6 months	81.0 ± 18.5	75.0 ± 23.6	0.724
KOOS Sport final follow-up	89.2 ± 7.4	92.7 ± 9.3	0.350
KOOS QOL preope	35.4 ± 29.0	54.2 ± 26.9	0.272
KOOS QOL post 6 months	70.0 ± 17.3	75.8 ± 21.8	0.622
KOOS QOL final follow-up	75.0 ± 11.9	85.8 ± 15.3	0.350

## Data Availability

The data used to support the findings of this study are available from the authors upon reasonable request.

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
