# Peer review of "Medial Patellofemoral Ligament Repair with Suture Tape Augmentation Can Yield Good Midterm Clinical Outcomes Regardless of Skeletal Maturity and Joint Laxity"

_biomimetics, 2025, doi:10.3390/biomimetics10010065_

Round 1
Reviewer 1 Report
Comments and Suggestions for Authors
Dear authors,
I had the privilege to review your manuscript. Please find my several comments below.
Abstract:
· Adequate, however the number of patients included is small, subgroup analysis are not possible with a meaningful information.
· KOOS scale might not be appropriate as a sole PROM for patellar instability
· You might want to give some numbers instead of the information of significant increase, since there is clinical relevant increase or mathematical increase.
Introduction
· Line 54: I don’t think you can talk of a native ligament, when it is a MPFL reconstruction with a tendon.
· Your introduction sounds like an advertisement rather than a scientific report, especially talking a lot about biomaterial. But these informations are not relevant for your outcome measures. Please try to be specific with your introduction, that must be in line with your study question and your outcome measures.
Methods
· How many patients did you have to exclude ? you mention 8 patients, but due to low FU-time however no other patient due to other exclusion criteria.
· An angle of 150° or more, was measured on x-ray or MRI? (you mention this latera on line 147 by conventional radiograph.. however the sulcus angle has not been validated on x-ray. Hence you should not use this).
· You mention you started by arthroscopy to assess trochlear and patellar dysplasia. Why is this assessment important? As an exclusion criterion you had a sulcus of 150° or less on imaging (I assume), or was it in arthroscopy ?
Results
· Adequate (tables are very long. I doubt that they are absolutely necessary)
Discussion
· adequate
Conclusions
· Adequate
Author Response
# Reviewer 1
Abstract:
Comment: Adequate, however the number of patients included is small, subgroup analysis is not possible with a meaningful information.
Response: We agree with the reviewer’s comment and have added the following text to the limitations section.
Lines 380--382: This study has several limitations. First, there was no control group, and the sample size was relatively small. A larger number of cases would be needed to make the current subgroup analysis more meaningful information.
Comment: KOOS scale might not be appropriate as a sole PROM for patellar instability
Response: Thank you for your comment. As noted, several studies suggest that the KOOS (Knee Injury and Osteoarthritis Outcome Score) may not be fully appropriate as the sole patient-reported outcome measure (PROM) for patellar instability. Crossley et al. developed a new subscale, the KOOS-PF (Knee Injury and Osteoarthritis Outcome Score for Patellofemoral Pain and Osteoarthritis), which is specifically designed to assess patellofemoral pain and osteoarthritis. This tool has demonstrated strong reliability, validity, and responsiveness [31] Additionally, a systematic review identified the KOOS-PF as the only PROM with sufficient content validity for patellofemoral pain, though the supporting evidence is of low-quality. [32]
In response to the reviewer’s comment, we have added the following sentences in the limitations section.
Furthermore, while we utilized the KOOS to evaluate clinical outcomes, it is possible that the KOOS may not be ideal as the sole PROM for assessing patellar instability.
The knee injury and osteoarthritis outcome score for patellofemoral pain and osteoarthritis (KOOS-PF) subscale was specifically developed specifically for patellofemoral pain and osteoarthritis and has shown good reliability, validity, and responsiveness. [31]A systematic review highlighted the KOOS-PF as the only PROM with sufficient content validity for patellofemoral pain, despite low-quality evidence supporting its use. [32] While the KOOS and its subscales remain valuable tools, condition-specific PROMs like the KOOS-PF may offer a more targeted assessment for patellofemoral issues and patellar instability. (refer Lines 307-313)
In addition, we have added the following references.
- Crossley, K.M., et al., The patellofemoral pain and osteoarthritis subscale of the KOOS (KOOS-PF): development and validation using the COSMIN checklist. Br J Sports Med, 2018. 52(17): p. 1130-1136.
- Hoglund, L.T., et al., Patient-Reported Outcome Measures for Adults and Adolescents with Patellofemoral Pain: A Systematic Review of Content Validity and Feasibility Using the COSMIN Methodology. J Orthop Sports Phys Ther, 2023. 53(1): p. 23-39.
Comment: You might want to give some numbers instead of the information of significant increase, since there is clinically relevant increase or mathematical increase.
Response: We agree with this comment. Accordingly, we revised our manuscript as follows.
Lines 20--23: PROs significantly improved from preoperatively to the final follow-up (IKDC score: 50.7 ± 26.6 vs. 88.8 ± 13.0, p < 0.001; KOOS: 68.8 ± 23.3 vs. 91.2 ± 8.4, P = 0.011) without reducing the patient’s activity level at the final follow-up (UCLA AS score: 7.9 ± 2.4 at preinjury vs. 7.9 ± 2.2 at the final follow-up, P = 0.655).
Introduction
Comment: Line 54: I don’t think you can talk of a native ligament, when it is a MPFL reconstruction with a tendon.
Response: We agree with the comment. We have deleted the following sentence from our manuscript.
The tape is supposed to act as a secondary stabilizer and to protect the native ligament during its biological healing.
Comment: Your introduction sounds like an advertisement rather than a scientific report, especially talking a lot about biomaterial. But this information is not relevant for your outcome measures. Please try to be specific with your introduction, that must be in line with your study question and your outcome measures.
Response: We agree with the reviewer’s comment. Accordingly, we have deleted the following sentences.
Lines 57--64: The development and application of bioinspired materials have revolutionized modern medicine by offering innovative solutions that mimic the properties of biological tissues. Polyethylene, a high-strength, biocompatible material widely used in biomedical applications, exemplifies this advancement. Its application in suture tape, as used in medial patellofemoral ligament (MPFL) repair, represents a significant step in enhancing clinical outcomes for patients with patellar instability [8] [9]. The durability and flexibility of polyethylene-based suture tape not only ensure structural integrity during repair but also address critical concerns regarding the longevity of clinical results.
Lines 71--78: This study aligns with the scope of biomimetics by evaluating the midterm clinical outcomes of MPFL repair with suture tape augmentation, bridging the gap between bioinspired materials and their long-term impact in biomedical applications [10]. By leveraging the properties of polyethylene to optimize ligament repair, this work provides valuable insights into the performance and sustainability of biomolecular materials in orthopedic surgery. Furthermore, our findings contribute to the growing body of knowledge on patient-reported outcomes and activity levels following advanced surgical interventions.
Methods
Comment: How many patients did you have to exclude? you mention 8 patients, but due to low FU-time however no other patient due to other exclusion criteria.
Response: Thank you for your comment. We excluded 8 patients because of a short follow-up period, and no other patient was excluded for other reasons. The inclusion and exclusion criteria were confusing, and we revised our manuscript as follows.
Lines 72--77: We retrospectively reviewed the clinical records of patients who underwent MPFL repair with suture tape augmentation between 2015 and 2020. The indications for the current procedure were patients who had at least one episode of patellar dislocation without trochlear dysplasia (sulcus angle > 150 degrees) or malalignment (tibial tubercle-trochlear groove (TT-TG) distance > 20 mm) because of the need for other procedures. The clinical records included patient demographics, radiographic variables and clinical outcomes. Patients with short follow-up periods (within 2 years) were excluded.
Comment: An angle of 150° or more, was measured on x-ray or MRI? (You mention this latera on line 147 by conventional radiograph. however, the sulcus angle has not been validated on x-ray. Hence you should not use this).
Response:
We appreciate the reviewer's comment regarding the measurement of the sulcus angle. To clarify, the sulcus angle is indeed a crucial parameter for evaluating patellofemoral congruence and potential instability. It has been validated for measurement on axial radiographs of the patellofemoral joint. Therefore, its application in this context is scientifically supported. We provide the following points for further clarification:
- Validation of Measurement on Radiographs:
- The sulcus angle was initially described for use in radiographs, particularly axial views of the patellofemoral joint. Subsequent studies have demonstrated its reliability and reproducibility across imaging modalities, including radiographs, CT, and MRI.
- A study published in BMC Musculoskeletal Disorders (2010) investigated the associations between indices of patellofemoral geometry, including the sulcus angle, and knee pain. This study highlighted the reliability of radiographic measurements in clinical evaluations (Ref: BMC Musculoskelet Disord. 2010 May 10; 11:87. doi: 10.1186/1471-2474-11-87).
- Furthermore, Severyns et al. (2023) emphasized the validity, reproducibility, and clinical significance of sulcus angle measurements on radiographs. Their findings confirmed that radiographic assessment remains a practical and valid method for this purpose (Ref: Knee Surg Relat Res. 2023 Jan 10; 35:1. doi: 10.1186/s43019-023-00175-5).
- Reproducibility and Reliability:
- Research has shown good intra- and interobserver agreement for sulcus angle measurements, underscoring its reproducibility across different evaluators and modalities. This agreement ensures that measurements are consistent, even when assessed by multiple observers.
- Consistency Across Imaging Modalities:
- While the sulcus angle was historically measured using radiographs, its application has been extended to CT and MRI, further broadening its utility. These modalities allow for a more comprehensive assessment of patellofemoral congruence and provide corroborative data to radiographic measurements.
Conclusion: We respectfully assert that the sulcus angle's measurement on radiographs is a validated and reliable approach supported by scientific literature. Therefore, its inclusion in our study methodology is appropriate and aligned with established practices in patellofemoral joint evaluation.
We have also added the following reference and the citation (Line 140)
Severyns, M., et al., Radiographic measurement of the congruence angle according to Merchant: validity, reproducibility, and limits. Knee Surg Relat Res, 2023. 35(1): p. 1.
We hope this clarification addresses the reviewer's concern. Thank you for the opportunity to improve our manuscript.

Comment: You mention you started by arthroscopy to assess trochlear and patellar dysplasia. Why is this assessment important? As an exclusion criterion you had a sulcus of 150° or less on imaging (I assume), or was it in arthroscopy?
Response: Thank you for your comment. As you mentioned, this was very confusing. In accordance with the reviewer’s comment, we have deleted the words ‘and dysplasia’ from our manuscript.
Before lines 84-85, standard diagnostic arthroscopy was performed to evaluate cartilage damage and dysplasia of the patella and trochlea.
After lines 84-85, standard diagnostic arthroscopy was performed to evaluate cartilage damage of the patella and trochlea.
Results·
Comment: Adequate (tables are very long. I doubt that they are necessary.)
Response: Thank you for your valuable suggestion. We would appreciate very much if we could keep these tables as they are since we believe this is the most reasonable way to show our results when comparing PROs.
Discussion
Comment: adequate
Response: Thank you for your comment.
Conclusions
Comment: Adequate
Response: Thank you for your comment.

Reviewer 2 Report
Comments and Suggestions for Authors
dear authors
fine paper
however as you stated , no control group ( which you must have " historically ?") as i read your expertise
see a few comments / suggestions in balloons in your text
language ok
refs iluustrations tables and legends are ok

Author Response
Comment: Line 2 this is not truly a repair! it is " replacement". we suggest changing the title accordingly
Response: Thank you for your suggestion. Since our procedure does not strictly remove the original ligament, we do not believe it can be considered a replacement. We would be the most grateful if we could keep the current title.
Comment: Line 102 when mentioning industry implants please give city and country please check throughout your paper
Response: We agree with the reviewer’s comment. Accordingly, we have added the following text to our manuscript.
Line 92: Overdrilled for SwiveLock (Arthrex, Naples, FL, USA) fixation (Figure 1A).
Line 100: suture tapes were fixed to the patella via SwiveLock (Arthrex, Naples, FL, USA) anchors…
Line 169-170: …the suture tapes and the SwiveLock (Arthrex, Naples, FL, USA) sutures via…
Comment: Line 222 new trauma?
Response: Thank you for your comment. One patient who experienced redislocation experienced a new trauma episode. We added the following text to our manuscript.
Lines 291--292: Redislocation was observed in 1 patient at 14 months after initial surgery because of a new trauma episode.
Comment: Line 226 there is no reason to doubt about this succes. but there is no control group to compare? do you have an historical comparing group available?
Response: Thank you for your comment. As you mentioned, the lack of a control group is one of the limitations of our study. It would be better to compare with isolated MPFL repair or MPFL reconstruction. We agree with the reviewer’s comment. Therefore, we mentioned this as a limitation. Please see lines 380--382.
Comment: Line 248 i very much doubt you can do that technically. especially as i see you incision sizes not allowing for finding back the original MPFL lig ends. please comment
Response: Thank you for your valuable feedback. We acknowledge the technical challenges associated with locating and preserving the original MPFL ligament ends, especially considering the small incision sizes. However, our surgical technique relies heavily on precise anatomical landmarks to ensure accurate identification and reattachment of the MPFL. To address your concern, we have revised the figure legend of Figure 1 as follows.
Lines 173--181: (A) Schöttle’s point was identified under fluoroscopic guidance, and an anchor was inserted precisely at the femoral insertion point. (B) Careful dissection through the minimal incision allowed visualization of the MPFL-capsule complex, and suture tape was passed through the MPFL substance via an Arthropierce device. (C) Two SwiveLock (Arthrex, Naples, FL, USA) anchors were placed at the 1:00 and 3:00 positions on the patella to cover the medial patellofemoral complex (MPFC) while maintaining isometry. (D) The MPFL-capsule complex was repaired via the lasso loop technique with suture tape and SwiveLock (Arthrex) anchors, ensuring proper tension and alignment within the trochlear groove.
Comment: Line 308 also no doubt here but your group is quite small to state this.
Response: Thank you for your comment. As you mentioned, the small sample size is one of the greatest limitations of our study. We acknowledge that future large-scale, multicenter studies are needed. We agree with the reviewer’s comment. Therefore, we mentioned this as a limitation. Please see lines 380--384.
